# Dynamic Oral Texture Properties of Selected Indigenous Complementary Porridges Used in African Communities

**DOI:** 10.3390/foods8060221

**Published:** 2019-06-21

**Authors:** James Makame, Tanita Cronje, Naushad M. Emmambux, Henriette De Kock

**Affiliations:** 1Department of Consumer and Food Sciences, University of Pretoria, Private Bag X20, Hatfield, Pretoria 0028, South Africa; naushad.emmambux@up.ac.za (N.M.E.); riette.dekock@up.ac.za (H.D.K.); 2Department of Statistics, University of Pretoria, Private Bag X20, Hatfield, Pretoria 0028, South Africa; tanita.cronje@up.ac.za

**Keywords:** oral processing, TCATA, texture, malnutrition, sensorimotor readiness, complementary porridge, infant

## Abstract

Child malnutrition remains a major public health problem in low-income African communities, caused by factors including the low nutritional value of indigenous/local complementary porridges (CP) fed to infants and young children. Most African children subsist on locally available starchy foods, whose oral texture is not well-characterized in relation to their sensorimotor readiness. The sensory quality of CP affects oral processing (OP) abilities in infants and young children. Unsuitable oral texture limits nutrient intake, leading to protein-energy malnutrition. The perception of the oral texture of selected African CPs (*n* = 13, Maize, Sorghum, Cassava, Orange-fleshed sweet potato (OFSP), Cowpea, and Bambara) was investigated by a trained temporal-check-all-that-apply (TCATA) panel (*n* = 10), alongside selected commercial porridges (*n* = 19). A simulated OP method (Up-Down mouth movements- munching) and a control method (lateral mouth movements- normal adult-like chewing) were used. TCATA results showed that Maize, Cassava, and Sorghum porridges were initially too thick, sticky, slimy, and pasty, and also at the end not easy to swallow even at low solids content—especially by the Up-Down method. These attributes make CPs difficult to ingest for infants given their limited OP abilities, thus, leading to limited nutrient intake, and this can contribute to malnutrition. Methods to improve the texture properties of indigenous CPs are needed to optimize infant nutrient intake.

## 1. Introduction

Food oral processing (OP), the manipulation and break down of food inside the mouth up to the moment of swallowing [1,2], plays a key and important role in sensory perception, consumer acceptance, and food intake [3]. It is a dynamic process; however, the scientific explanations regarding the sensory quality of baby food is lacking [4], particularly the effects of interactions between complementary porridge (CP)’s texture properties and the oral physiology of infants. Oral texture perception remains poorly understood, despite it being a key driver of food acceptance or rejection [5]. Most infants in Africa are nourished on low-cost complementary porridges (CPs) prepared from starchy plant materials (cereals, roots, tubers, and legumes) [6,7]. Such porridges are often thick even at low (about 8–10%) solids content [8,9,10]. In infants and young children, physiological capacity (chewing, salivation, and digestion), sensory quality and oral motor skills are important determinants of food choice [11]. Food texture is a perception arising from the interactions of the food physical structure with mechanoreceptors in the oral cavity [5]. Inappropriate porridge viscosity may compromise nutrient intake and lead to child malnutrition, a major public health problem, especially in low- to middle-income countries [12].

Children of different ages have different OP abilities to successfully chew and swallow foods of different physical forms [13]. The process of bolus formation is under the coordinated action of mastication (reduction of food in particles), salivation (lubrication of particles), and tongue movements (agglomeration of particles with saliva and swallowing) [1] and depends, thus, mainly on the development of the infant masticatory apparatus. For ingestion and break down of solid foods, children need to acquire specific feeding skills which require more effort than the oral manipulation of liquid milk [14]. The acceptance of food with a given texture (in this context defined as the infant’s ability to swallow the food [11]) strongly depends on the acquisition of feeding skills, which can develop differently in children of the same age [15]. At 12 months, munching/chewing behavior is well-established and continues to develop and optimize by 2–3 years [16]. However, the age of chewing maturation (i.e., transition of up and down movements of the jaw to rotary movements) remains not so clear and is estimated to be later than 3 years [14]. Good quality CPs must have low viscosity, high nutrient density, appropriate texture, and a consistency that allows for easy consumption by infants and young children [17,18]. At the beginning of complementary feeding (4–6 months), infants prefer soft and smooth textured foods as they require limited oral manipulation before being swallowed [11]. Commercial infant porridges are considered to have the optimal quality, but they are too expensive for many poor families [19].

Bolus flow and ease of swallowing depend on the rheology of the bolus and the oral physiological conditions of the consumer [20,21]. Infants have limited dentition, weak masticatory muscles, and reduced tongue or pharyngeal muscle strength [22]. For safe and comfortable consumption, the porridge consistency should be matched with the child’s oromotor readiness [23]. At present, there is paucity of research on the relationship between the in-mouth perceived texture of indigenous CPs and the sensorimotor development (oromotor readiness) of infants (6–12 months) and young children (13–24 months). Yet, infants and toddlers present a challenge to sensory and consumer researchers because of their inability to communicate verbally, limited cognitive abilities, and very low attention span [24,25]. Sensory testing with infants and young children, therefore, has often employed indirect approaches. As an example, descriptive sensory profiling has been used to evaluate the sensory quality of baby foods (purees) [4]. 

For preference evaluation, parents’ liking is important in deciding if a given CP would be suitable for their infants [26]. The primary caretaker (typically the mother) interprets the behavior of the child during food tasting and rates acceptance on a hedonic scale [11,27,28]. The adult also tastes and rate the samples after the child, providing a control and confirmation of the acceptability of the samples [25]. In most studies, mothers are often asked to report on the presence/absence of positive and negative behaviors and on infant’s food liking during feeding. Alternative testing approaches employed include parents completing an Infant Behavior Questionnaire and rating of videotapes of infants’ facial reactions to foods rated [29].

Data from rheological studies (not included in this paper) have shown that indigenous porridge samples at very low solids content (Maize 8.1%, Sorghum 8.4%, Cassava 6.4%, Bambara groundnut 10.7%, and Cowpea 10.1%) exceeded the recommended CP viscosity limit (3 Pa·s at 40 °C and shear rate of 50 s^−1^, [30,31]) for infants and children below 3 years of age. When starch is heated in water, it swells, gelatinizes, and pastes to form a thick gruel [8,9,10]. Infants and young children have difficulty to consume and swallow a viscous porridge due to their limited oromotor capacity [32]. The thickness or viscosity of shear-thinning foods is perceived by mechanoreceptors in the mouth, and oral thickness perception depends on the in-mouth shear stress applied and the resultant shear rate [33]. At the critical solids concentration (c*), the gelatinized, amorphous random starch polymer coils come in contact with one another, eventually overlapping at the entanglement concentration (c_e_) [34]. Weight for weight, polymers with larger molecules display more effective molecular entanglements [35,36], and are generally perceived as more viscous and thick compared to those with smaller moleules. High viscosity in CP elicits high lingual swallowing pressure [37]. Thicker and harder foods are eaten at a slower rate, often requiring smaller bite sizes and more chewing time in the mouth before swallowing compared to softer foods [38]. The formation of a bolus that can be safely swallowed is a complex oral process [11], and infants and young children have limited oral capacity to perform this process. 

The aim of this study was to characterize the texture of selected indigenous CPs typically used for feeding infants and young children aged 6–24 months in African communities during OP (therefore dynamic), in order to make recommendations for optimizing their oral texture to improve nutrient intake. A trained sensory panel consisting of adults was used because infants are not capable of carrying out evaluation instructions and tasks expected in descriptive sensory evaluation. To understand the temporal in-mouth textural nuances, the design applied two different OP methods: a novel procedure (the Up-Down mouth movements- munching) that mimics how infants with limited OP abilities process food, and a control method (chewing with lateral mouth movements) representing normal adult OP.

## 2. Materials and Methods

### 2.1. Samples and Sample Preparation

Table 1 shows the descriptions of indigenous and commercial CP samples used in the study. To make flour, Bambara groundnut (*Vigna subterranean* (L.) Verdc) and Cowpea *(Vigna unguiculata*) seeds were first decorticated using a Tangential Abrasive Dehulling Device (TADD, Venables Machine Works, Saskatoon, Canada) and milled to <250 μm particle size flour (Laboratory mill 3100, Perten Instruments, Hägersten, Sweden). All indigenous CP samples were prepared as described by [39] with adaptations. A specific amount of flour was measured into an aluminum pot following Table 1, and 250 mL of cold water was added while stirring to form a uniform slurry. The flour quantities for each treatment represent the solids % determined from rheological experiments, that give the recommended porridge viscosity limit of 3 Pa·s (at 40 °C and shear rate of 50 s^−1^). The pot was placed over a hot plate, and the remaining quantity of boiling water was slowly added to the slurry while continuously stirring. Once the mixture began to boil, the timer was started, and the porridge was cooked for 7 min with continuous stirring. Porridge samples were transferred to appropriately labeled containers placed over a water-bath maintained at 55 °C until serving. Commercial porridges were prepared, according to the manufacturers’ instructions, by mixing with water or milk depending on the product. The processing techniques for commercial porridges allow them to remain thin at a higher solids % compared to indigenous porridges.

### 2.2. Sensory Analysis

The sensory analysis of the porridges was conducted in individual booths under white light and standardized conditions at the University of Pretoria Sensory Evaluation Laboratory. The use of human subjects in the study was approved by the Faculty of Natural and Agricultural Sciences Ethics Review Committee at the University of Pretoria (EC 180000086). Each participant signed a consent form prior to taking part in the study.

Ten assessors (3 males and 7 females, aged 22–27 years) were selected and trained on the Temporal Check-All-That-Apply (TCATA) evaluation method in three sessions of 2 h each, according to the guidelines of the International Organization for Standardization (ISO) standard 8586:2012 [40]. Prior to training, panelists were screened for interest, availability, general health status, and product discrimination abilities. With TCATA, panelists taste the samples and select all the sensory attributes they perceived at each moment of the evaluation [41]. They are allowed to check several attributes, which enables them to describe sensory characteristics that are simultaneously perceived [42]. During training, the assessors familiarized themselves with the oral texture of the CPs, discussed, and agreed on 14 attributes (Table 2). The panel was also trained on the evaluation protocol and use of the data acquisition software, Compusense Cloud version 7.8.2 (Compusense Inc., Guelph, ON, Canada).

TCATA has often been used with a large number of consumers in evaluating different products [43,44]. However, coupled with training, fewer panelists (10–15) have also previously been used for the temporal profiling of products based on 8–10 attributes [41,45]. The current study used 10 panelists well trained in carrying out the TCATA task and a list of 14 attributes. Temporal methods are more cognitively demanding and usually rely on shorter lists [46,47]. 

Thirty-two CPs (Table 1) were evaluated in duplicate, using two different OP methods, over 7 days with a maximum of ten CPs per day. The CPs were first evaluated using the Up-Down method that mimics feeding in infants and young children with limited OP ability. Assessors moved their mouths only up and down, avoiding sideways movements and ensuring limited tongue movement. Babies initially use immature feeding skills characterized by the up and down movements of the jaws, eventually transitioning to mature feeding skills defined by rotary jaw movements, which facilitate efficient chewing [14,23]. In the second method (experienced adult chewing called Normal), panelists orally processed the food in a normal adult way involving the lateral mouth and tongue movements, applying oral shear and chewing where necessary. The Up-Down method was always used first before the Normal method during the evaluation sessions because the former required more conscious procedural effort due to its artificial nature compared to normal adult oral processing.

All attributes were presented in a three-column format on the computer screen. The order position of attributes on the TCATA list was randomized across assessors, but the list order remained consistent for a given assessor across all samples [52]. Porridge samples (±20 g) were presented to assessors monadically at 40 °C in glass ramekins covered with aluminum foil following a random balanced order. The evaluation instructions requested the panelists to select all terms on the TCATA list (Table 2) that described the sensations experienced at a given time of evaluation (measurements per second). Concurrently, they had to deselect the terms that were no longer relevant to describe the sensation of a given sample at that moment during the evaluation. To begin the evaluation, assessors took a spoonful (±5 g) of porridge into their mouth, clicked “start” on the computer screen (*t* = 0 s), and proceeded according to the instructions. After 20 s, they were prompted to swallow the sample and continued to note the sensations until the end of the evaluation. The evaluation duration (30 s) was established through an iterative process with assessors during training. It was an estimate of the time required for assessors to orally process a spoonful of porridge from intake to swallow while eating like a baby (munching). A 1 min break was enforced between samples for assessors to rinse their palate with deionized water. 

### 2.3. Data Analysis

Attribute citations proportions were calculated using a procedure described by [53], as the percentage of assessors who selected an attribute (‘1’) at any given moment (every 1 s) during the evaluation period. TCATA curves were plotted using statistical software R [54], package tempR (version 0.9.9.15.) [55], and the lines smoothed by the cubic smoothing spline function to reduce noise in the data. For each attribute, reference lines per treatment at every 1 s during the evaluation period were calculated according to [56], as the average across all other CPs excluding the one that this average is contrasted with. Principal Component Analysis (PCA)—a multivariate data analysis method for visualization of correlations between multiple quantitative observations and variables [57], was used to produce PCA product trajectories (biplot). Visualizing observations in a 2- or 3-dimensional space permits identification of uniform or atypical groups of observations. PCA can be considered as a projection method, which projects observations from a *p*-dimensional space with p variables to a *k*-dimensional space (where *k* < *p*) so as to conserve the maximum amount of information from the initial dimensions [57].

For the average citation proportions, TCATA data were divided into 3 equal time slices of 10 s each (Initial: 1–10 s; Middle: 11–20 s; End: 21–30 s), and the mean values were obtained for each attribute at each time slice, as the proportion of the 10 s evaluation time that the attribute was selected. For example, if an assessor selected chewy for a duration of 8 s and creamy for a duration of 2 s, then the citation proportions for chewy would be 8/10 = 0.8, and for creamy would be 2/10 = 0.2. Data were analyzed using a mixed effects ANOVA model:AUC = Porridge Sample + OP Method + Porridge Sample × OP Method + Assessor,(1)
where AUC refers to the area under the citation proportion by time (s) curve (average citations proportions) for each attribute in each treatment. The Sample, Method, and their two-way interaction effects were the fixed factors, whereas Assessors was a random effect. Multiple comparisons were done using Fischer’s Least Significant Difference (LSD) test to detect which sample pairs were significantly different. XLStat software (version 2019.1.2) (Addinsoft, Long Island, New York, NY, USA) [58] was used for the analysis. Citation proportions (range: 0 to 1) correspond to the AUC in Time Intensity studies [48]. In order to test if the porridge samples differed significantly with respect to presence or absence of the 14 specific oral texture attributes, the data, pooled over OP methods and replications, were submitted to Cochran’s Test, as described by [45]. Cochran’s Q is used for testing *k* = 2 or more matched sets, where a binary response (e.g., 0 or 1) is recorded from each category within each subject. Cochran’s Q test tests the null hypothesis that the proportion of “successes” is the same in all groups versus the alternative hypothesis that the proportion is different in at least one of the groups. For pairwise differences between each treatment within a time-segment, Marascuilo’s Test was used [59]. The Cochran’s Q test followed by Marascuilo’s test was done in XLSTAT software (version 2019.1.2) (Addinsoft, Long Island, New York, NY, USA) [58]. 

## 3. Results

### 3.1. Dynamic Oral Texture: TCATA Product Profiles

Figure 1A–F shows TCATA curves for the three most common types of African CP (Maize, Sorghum, and Cassava, all 10% solids). When evaluated by the Normal OP method (Figure 1A,C,E), these porridges were perceived (*p* < 0.05) as more thick or too thick, more sticky, and less creamy during the initial (1–10 s) and middle (11–20 s) phases in comparison with the mean (dotted lines) of all other samples. In the Up-Down method, maize porridge (Figure 1B) was thicker (*p* < 0.05) for a much longer time period, with much higher citation proportions than in the Normal OP method (Figure 1C). The panel also perceived Cassava porridge as thick, too thick, sticky, slimy, and not creamy by both OP methods (Figure 1E,F). In the Up-Down method (Figure 1F), however, the Cassava porridge attributes were more prevalent and more persistent, being further described as significantly not thin (between 10 s and 15 s) and less easy to swallow compared to the rest of the samples. 

The Cochran’s Q test was carried out on the TCATA data collected during the 30 s evaluation period for the indigenous CPs and a commercial reference (Table 1 sample A1). The frequency of perception (citation proportion) for attributes used differed in time (*p* ≤ 0.05) as a function of Samples (Table 3). The 6th, 16th, and 26th s time moments were taken to represent three phases of OP during the evaluation. Initially and mid-way during OP, assessors described Maize and Sorghum porridges (8–10% solids) and Cassava 10% as thick, and even too thick, relative to the other CPs (*p* ≤ 0.05). Cassava CPs (6 and 10% solids) were also characterized as significantly (*p* < 0.05) sticky and pasty and together with Bambara groundnut and Cowpea (all perceived as slimy *p* ≤ 0.05) when compared with the other porridges. CP attribute differences declined as OP progressed, meaning that the CPs became more similar towards the end of OP. During swallowing, more panelists perceived Cassava 10% as thick, sticky, pasty, and slimy compared to the other porridges (*p* ≤ 0.05). The slimy texture was more frequently perceived in all legume porridges (Bambara groundnut and Cowpea) but not in the cereal and orange-fleshed sweet potato (OFSP) based porridges.

When applying a 2-way mixed model ANOVA to evaluate the effect of Sample and OP method (Table 4), similar results were noted. Maize and Sorghum (8–10% solids) and Cassava (10% solids) were characterized by the panel as significantly thick or too thick, sticky, slimy, and pasty compared to the rest of the samples, in all three time-slices (only results for initial and end time-slices are shown in Table 4). The Up-Down OP method gave significantly higher thick and too thick citation proportions compared to the Normal OP method for Maize (8–10% solids), Sorghum (8–10% solids), and Cassava (10% solids) porridges. As in Cochran’s Q test, slimy texture during swallowing was more perceived in cassava and leguminous CPs, with Cassava 10% described as not easy to swallow by a higher proportion of panelists.

### 3.2. Dynamic Oral Processing Trajectories for Selected Indigenous/Local CPs and a Commercial Porridge Reference

The correlation between the CPs and attribute changes over the 30 s OP duration (both Normal and Up-Down methods) was explored graphically via PCA product trajectory biplots on three dimensions (D1 to D3) (Figure 2). 

The first three dimensions (D1 to D3) in the normal OP method explained 79 % of the total inertia, while for the Up-Down OP method, these explained 75% of the model information. For both methods, during the evaluation, D1 (Figure 2A,C) explains the early CP differences due to attributes thick, slimy, sticky (and too thick in Up-Down method) in contrast to differences in watery and easy to swallow perceived towards the end of OP. Soft, grainy, smooth, and thin became relevant differentiation attributes midway in the evaluation. In both methods, D2 is associated with CP differences described as smooth, thin, watery, and creamy (Figure 2A,C). It is, however, clear that the temporal distribution and trajectory of CPs is different in D2 for the two OP methods. This implies that the CPs are perceived differently in time, when processed by Up-Down or Normal methods. With the Up-Down method (Figure 2C), the commercial porridge (A1) is clearly distinguished from the other CPs and, particularly, Cassava (C). With Normal OP (Figure 2A), the commercial porridge was not distinguished in D2 from most of the indigenous CPs, while OFSP (O), Cassava (C), and Maize porridges were more closely grouped. However, while all CPs were watery and easy to swallow, at the end of OP, in the Normal method, there was more variation with regards to these attributes amongst CPs in the Up-Down method (Figure 2C). Cassava was the least watery and least easy to swallow. Differences in the perception of the CPs as a function of the two OP methods are also explained in the D3 axes of the respective maps (Figure 2B,D). With the Normal method (Figure 2B), D3 separates CPs that are easy to swallow, smooth, soft, and slimy at the top from those that are grainy and to a lesser extent watery at the bottom of the plot (notably OFSP, O). With the Up-Down method (Figure 2D), slimy at the top is the main distinguishing attribute, particularly, contrasting the texture of mainly Cassava CPs to the grainy and watery character of OFSP (O) at the bottom of the plot.

## 4. Discussion

The early detection of viscosity-related attributes (too thick, thick, sticky, slimy, pasty) during OP of CPs was consistent with the generally known evolution of sensory perception during OP of semi-solid foods, as reported by [60]. The attributes thick, too thick, sticky, pasty and slimy, and less easy to swallow characterized the most common indigenous African CPs (Maize, Cassava, and Sorghum, 6–10% solids) early in OP. Similarly, tracking the changes in the oral texture of soft and semi-solid foods, [61] found that bulk-related attributes, such as thickness, dominate the initial phase of OP.

The characterization of Cassava, Maize, and Sorghum CPs (all 10% solids) as sticky, thick, too thick, slimy, and pasty, especially when eaten using the Up-Down method, has important implications for infant feeding. Children generally dislike sticky and slimy textures because of their lack of control over these texture attributes in the mouth [62]. A study [63] on the temporal texture quality of two soft cereal products of different composition (sponge-cake and brioche) for the elderly found that oral eating quality was negatively correlated with perceived stickiness and pastiness, but positively correlated with perceived easiness to eat, easy to chew, and easy to swallow the food bolus. The authors reported that sticky and pasty attributes were perceived and characterized as unpleasant [63]. According to [64], stickiness in the mouth and cohesiveness are the most important textural attributes to control in thin porridge. The developmental differences in oral physiology may impact the infants and young children’s ability to orally process thick and sticky indigenous CPs. The age of a child impacts chewing and mastication abilities [65]. However, [14] observed that starting from 6 months onwards, acceptance of sticky textures was increasing with increasing infant age most likely due to familiarity. Eating skills develop gradually from sucking to munching and then chewing (a more complex rotary pattern) [15]. The capacity to transform food into a safely swallow-able bolus influences food acceptability and nutrient intake, yet it is severely limited in infants and young children [11]. For easy swallowing, the food bolus should be readily deformable and flowable [66]. The eruption of front teeth typically between 6 and 8 months and the distal ones from 12–24 months of age improves the ability to break down more challenging foods only later in infancy [67]. The change from the munching pattern to a more adapted, rotary chewing pattern occurs after at least 12 months of age [67,68]. This is because, in infants, milk and liquid foods are delivered directly to the posterior of the oral cavity during sucking and suckling- the optimal spot for swallow reflex initiation [67]. However, for handling solids, the infant must accept the food into the front of the mouth, masticate it, and then actively (with energy expenditure) use the tongue to transport the bolus to the posterior of the oral cavity for the swallow reflex to be triggered [69]. Mastication of solids-like, more challenging foods requires different chewing movements in bolus preparation [70], which are not present in early infants. 

Cassava and Maize porridge (at 10% solids) were described as the least easy to swallow and still thick (21–30 s) in the Up-Down OP method; observations which were absent when these porridges were consumed in the Normal OP method. Although the oral physiology of infants is quite different from that of adults [71], in a study involving young adults (mean age 26.5 years) with a high number of functional dental units, and some elders (mean age 67.2 years) with varying numbers of opposing post-canine teeth pairs (i.e., functional units) it was found that the same amount of work is needed to transform food from its initial form to an easy to swallow bolus regardless of age. The lingual propulsive force in adults is thought to be the main driving force for bolus flow [72], making OP much easier in adults than in infants. Reduced dentition as is the case in infants compromises masticatory efficiency, as well as the tongue’s capability in positioning food, and this reduces the efficiency of food oral break down [73,74]. It is acknowledged that young children and adults differ in their oral processing abilities, and that the adults’ usage of different oral processing apparatus, saliva secretion, etc. cannot fully explain the texture perception and eating behavior of small children.

Cassava porridges (6–10% solids) were described as smooth during the initial OP phase and easy to swallow in the normal OP method, but thick, sticky, and less easy to swallow with the Up-Down OP method. The capacity of the Up-Down OP method to detect sensory differences at relatively high viscosity levels confirmed a proposition [75], that low shear viscosity is more relevant in differentiating thickness perception in fluid foods. The Up-Down method employed low shear viscosity, making handling and break down of thick and sticky porridges more problematic, while shear was higher with the Normal OP method, enabling greater intraoral food-mixing ability. In a research study on the dynamic texture break down of some soft foods (caramel), [76] noted that an increase in stickiness only led to an increase in the total amount of muscle used but to a slower masticatory process with larger opening and closing strokes. In infants, the more advanced motor skills for handling semisolid foods only appear between 9–12 months, followed by molars at 12–18 months [77]. As clinically noted, when children do not have the required OP abilities to break down foods, they often hold them in the mouth to soften with saliva and/or attempt to swallow the pieces whole, which increases the risk of choking [67,69,78]. The α-amylase and the lingual lipase enzymes in saliva are thought to digest starch and lipids respectively, partially contributing to a decrease in the perceived thickness of fluids [79]. However, in this study, saliva may not have influenced the differences in texture across OP methods, since the panel was constant.

Maize porridge (10% solids) in the Up-Down method was pasty midway during OP, while soft and easy to swallow by the Normal OP method at 8 % solids. The apparent more pleasant oral texture perception in the Normal OP method may be explained by the greater efficiency of use of dentition and other developed oral structures, such as the tongue, palate and jaws. Teeth action improves oro-tactile sensitivity and assists in cutting and grinding of more textured foods [80]. The tongue has a fundamental role in bolus containment and propulsion [81]. Fluid food thickness affects the chewing rate and muscular work [82]. A thick food bolus is difficult to deform and flow, as swallowing muscles may have to work much harder and much longer to generate enough oral pressure/stress to transform the bolus into appropriate flow-ability for swallowing [66]. The clear shift in oral texture from thick, sticky, and too thick, to thin and watery in the Normal OP method compared to the Up-Down method was probably due to a high shear rate in the Normal OP method because of a more complete (unrestricted) and effective use of the fully developed oral physiology. In a study on infant formulas, [83] reported a similar decrease in viscosity with increasing shear load (non-Newtonian shear thinning behavior). The oral shear rates in infants are extremely low due to the absence of lateral jaw movements. When a bolus is easily deformable and flowable, it can be swallowed comfortably with minimal oral effort. A limitation of the study is that the panel always applied the Up-Down method first followed by the Normal OP method, which could potentially affect results through possible order effects.

The relative increase in the initial perceptions of thick, too thick, sticky and pastiness of CPs with increasing solids content show a progression towards a more solid food state, which would be more challenging for infants to orally process. In infants whose OP is limited, this may lead to increased difficulty in food oral break down and oral manipulation. Food structure determines, to a large extent, how fast the food falls apart in the mouth, which influences the total chewing time and number of chews [84]. The impact of food composition on texture perception originate from differences in microstructural and physical-chemical properties [60]. The bulk volume of indigenous porridges results from gelatinized starch [85]. Consumer perception of the texture of porridge is influenced by processing technique and solids concentration [86]. For commercial CPs, processing steps, such as hydrolysis, dextrinization, and pre-gelatinization, contribute to pleasant oral sensory texture by breaking down complex and long chains of food biopolymers into short-chain molecules, lowering thickness and stickiness of porridges [83,87]. Foods are processed differently in the mouth, depending on their physical-chemical and mechanical properties [84]. As observed by [88], high bolus viscosity increases submental electromyographic (EMG) activity, indicating the use of additional muscular force during swallowing as viscosity increases. [89] reported that compromised OP capabilities (e.g., during transporting food to the mouth, opening and closing the mouth, and swallowing) is closely associated with a high level of eating difficulty, low energy intake, and malnutrition. 

Food composition and structure also influence mastication (number of chews) and salivation [90]. Ref. [91], studying the sensory properties of a variety of foods showed that the stress applied during consumption depends on the viscosity of the food. Low viscosity foods were observed to be associated with minimum stress and increasing rate of deformation, while for high viscosity foods, the deformation rate was maintained as the stress was increased [84]. 

The effect of the OP method, porridge type, and the temporal nature of OP (i.e., the evolution of oral texture attributes over time) on the in-mouth texture perceptions of CP samples was demonstrated. Food texture perception is a highly dynamic process that depends on the constant manipulations and transformations of foods in the oral cavity [60]. In both OP methods, the initial texture of Maize, Cassava, and Sorghum CP (10%) was described as thick and/or too thick, sticky, slimy, progressing to soft, smooth, watery, then easy to swallow towards the end of OP. This is partly due to the shear-thinning behavior of the CPs, in addition to food oral breakdown during OP. At very low shear rates (zero-shear viscosity η0), polymer suspensions are entangled many times, showing visco-elastic behavior arising from the balance between molecular disengagements and elastic recoil when an initial shear force is applied [35]. Each polymer chain assumes a spherical shape, entangled many times with neighboring macromolecules, leading to a higher viscosity at rest [36], often perceived as initial thickness and stickiness. When a shear load is applied during chewing, food molecules disentangle to a certain extent and gets aligned in the shear direction, and agglomerates disintegrate releasing bound liquid to flow again [35]. Together with the possible dilution and hydrolysis effects of salivary compenents, these events reduce porridge viscosity as OP progresses, which is perceived as a thin and watery consistency towards swallowing.

According to [92], the attribute “soft” when used to describe a food during OP implies pleasantness, and “easy to swallow” denotes a pleasant feeling as the bolus pass through the throat. “Viscous (thick)” and “sticky” imply unpleasantness of a material that adheres to or entangles on the eating utensils or teeth and is difficult to remove. OP was more complete in the Normal method as all samples achieved a watery and easy to swallow state, while in the Up-Down method, the samples achieved varying degrees of OP at the 30 s end-point, with Cassava (10% solids) the least easy to swallow. This correlated strongly with its perceived stickiness, pastiness, and sliminess prior to swallowing. 

The reference porridge (A1) was characterized as creamy, smooth, and soft towards the end of OP. The feeling of creaminess is associated with the lubrication properties of the oil droplets between the tongue and the palate [84]. Smoothness is a complex tactile sensation implying the absence of graininess [93]. From an oral motor development perspective, optimal complementary foods should be well suited to infants’ chewing and swallowing abilities in order to provide a pleasant early feeding experience. Yet that seems to not be the case with common indigenous and locally available CPs. An immediate effect of eating difficulty is reduced food intake, increasing the risk of malnutrition [94,95]. 

## 5. Conclusions

This study applied a trained adult sensory panel to gain insights into the temporal oral texture characteristics of indigenous porridges for infants and young children. The oral texture sensory perceptions of porridge samples were different depending on the OP method used. Indigenous CPs were thick, sticky, pasty, and slimy even at very low solids content, making the porridges potentially difficult to process, unpleasant, and not easy to swallow. The Up-Down OP method that mimics the restricted oral processing abilities of infants and young children leads to more enhanced perceptions of the thick, too thick, sticky, slimy, pasty, and difficulty to swallow attributes. This could ultimately limit food and nutrient intake, perpetuating protein and energy malnutrition in infants that rely on these food types. OFSP porridge had a satisfactory oral texture at its highest solids content, comparable to a commercial reference (A1). Parents and caregivers are advised to consider the use of OFSP flour in composite with a legume (e.g., Cowpea or Bambara) for the preparation of CPs with relatively high solids content, suitable oral texture, and nutritive quality. Simple traditional approaches for reducing the viscosity of indigenous CPs at home (e.g., malting, fermentation, souring) need consideration by primary caregivers. This study provides scientific insight for baby foods manufacturers on the OP characteristics of complementary foods for infants and young children in African communities. Smart tech innovations for processing indigenous flours to give CP an optimal oral texture at much higher solids content for improved infant nutrient intake, are required. Moreover, further research is needed to explore the dynamic sensory interplay between bolus properties of baby foods and the oro-tactile phenomena (tongue coordination, mastication, and lubrication) in infants and young children, which, at present, is not well understood. 

## Figures and Tables

**Figure 1 foods-08-00221-f001:**
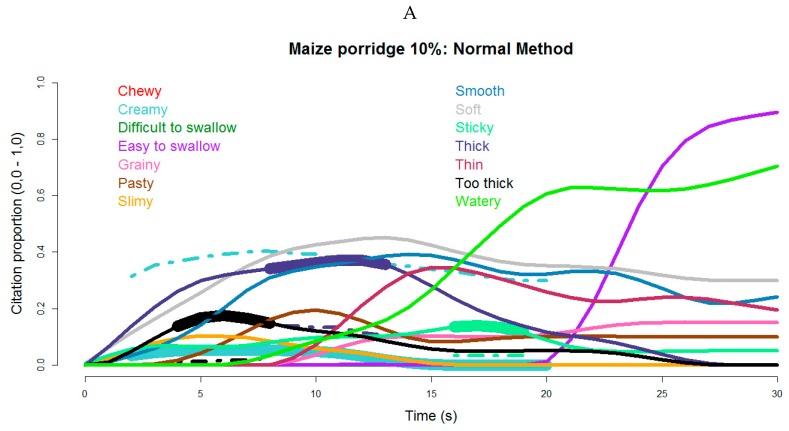
Temporal Check-All-That-Apply (TCATA) texture attribute curves for maize, sorghum, and cassava complimentary porridges (CPs) (10% solids). Attribute reference lines (represented as dotted lines in the figures) are shown only during periods of significant differences (*p* ≤ 0.05) in citation proportion for that porridge compared to the mean of all other CPs. Significant reference line segments are contrasted with highlighted thicker sections of attribute curves for convenient visualization. The letter **A**–**F** represent maize, sorghum and cassava CPs evaluated by the Normal and the Up-Down OP method respectively.

**Figure 2 foods-08-00221-f002:**
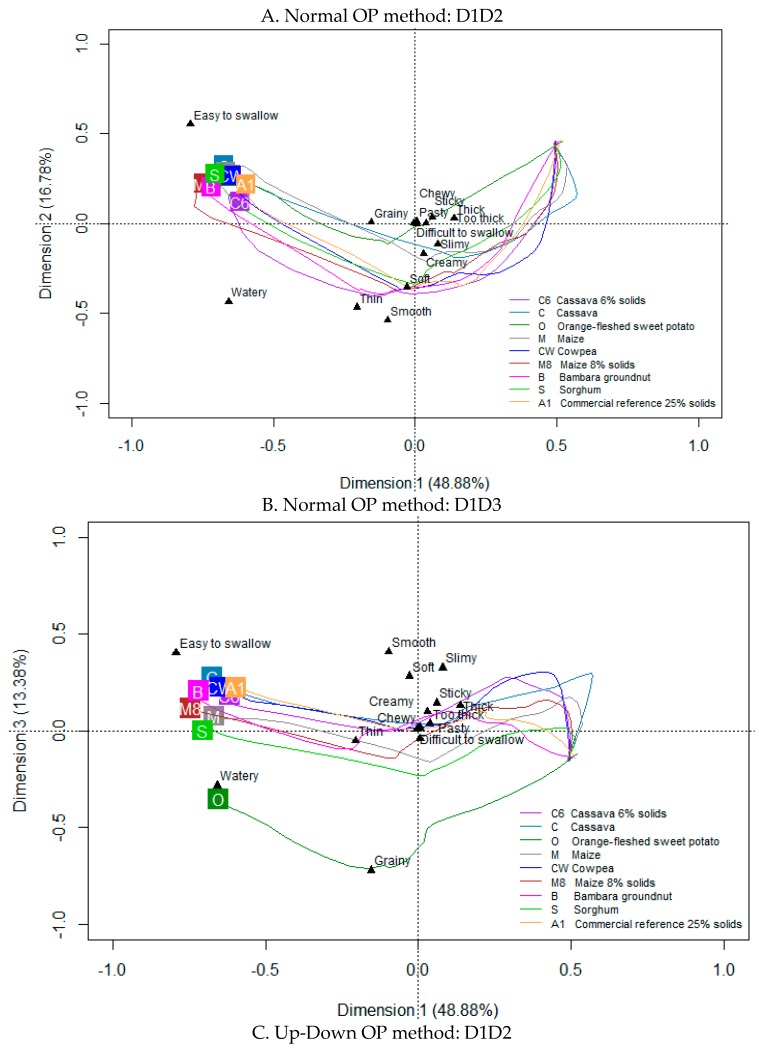
Smoothed Principal Component Analysis (PCA) product trajectory biplots show the evolution of attributes during oral processing for nine complimentary porridges (CPs) at 10% solids unless otherwise specified.

**Table 1 foods-08-00221-t001:** Description of complementary porridges (CPs) evaluated for oral texture by the trained TCATA sensory panel.

**Porridge Indigenous/Local**	**Flour (g)**	**Water (g)**	**Solids (%) #**	**Description and Source**
Maize	40	960	4	Super maize meal (commercially processed) from the local supermarket (Pretoria, RSA)
	80	920.0	8	
	100	900	10	
Sorghum	40	960	4	Super mabela flour (commercially processed) from local supermarket (Pretoria, RSA)
	80	920	8	
	100	900	10	
Bambara	100	900	10	Dry Seeds, cream cultivar, Mbare Produce market (Harare, Zimbabwe)
Cowpea	100	900	10	Commercial seeds, Agrinawa cultivar Agricol (Pty) Ltd. (Pretoria, RSA)
Cassava	40	960	4	High-quality cassava (84.4% starch), Thai Farm International (Ogun, Nigeria)
	60	940	6	
	100	900	10	
OFSP (Orange-fleshed sweet potato)	100	900	10	Dried with electric dryer (60 °C, 6–8 h), Exilite 499 cc (Tzaneen, Limpopo, RSA)
	160	840	16	
**Commercial Porridges (Code)**	**Age (Months)**	**Flour:Liquid (g:mL)**	**Solids (%)**	**Description/Manufacturers Instructions Guide** **
A1-Reference	6 to 24	50:150	25.0	Enzyme-hydrolyzed cereal (maize 62%), add water
A2	6 to 24	50:150	25.0	Enzyme-hydrolyzed cereal (rice 63%), add water
A3	6 to 24	50:150	25.0	Enzyme-hydrolyzed cereal (wheat 61%), add water
C2	6 to 24	50:140	26.3	Oat flakes 32%, add water
F1	6 to 8	45:150	23.1	Enzyme-hydrolyzed cereal (wheat 51%), add water
	9 to 12	67:200	25.1	
	13 to 36	80:250	24.2	
F2	6 to 8	35:150	18.9	Enzyme-hydrolyzed cereal (rice 51%), add water
	9 to 12	60:200	23.1	
	13 to 36	75:250	23.1	
F3	9 to 12	67:200	25.1	Enzyme-hydrolyzed cereal (wheat, rice, corn, rye, barley 54%), add water
	13 to 36	80:250	24.2	
F4	9 to 12	67:200	25.1	Enzyme-hydrolyzed cereal (wheat, rice, corn, rye, barley 43%), add water
	13 to 36	80:250	24.2	
B1	6 to 24	50:160	35.1	Enzyme-hydrolyzed cereal (maize), add water
B2	6 to 24	50:160	35.1	Enzyme-hydrolyzed cereal (wheat), add water
C1	6 to 24	20:170	24.8	Whole oat flour 70%, banana flakes 30%, add milk
D	6 to 36	20:140	26.3	Maize flour minimum 86%, add milk
E1	6 to 12	25:200	26.7	Maize meal flour, 3 min cook with milk *** Cooking loss of 5%
	13 to 36	35:280	25.8	
E2	6 to 12	25:125	30.0	Sorghum flour (minimum 89%), add milk
	13 to 36	35:190	29.1	
G	13 to 36	20:80	32.8	Wheat flour, maize flour, soy flour, add milk

* NAN Optipro milk (Formula 2 for 6–12 and Formula 3 for 13–24 months, respectively) was prepared as per manufacturer (mixing 32 g milk powder with 200 mL pre-boiled luke-warm water) to give a 16% solids content milk. The “add-water” commercial samples contain whole or skim milk (23–40%). # determined from rheological experiments, such that the flour % in water gives the recommended cooked porridge viscosity ≤3 Pa·s (at 40 °C and shear rate of 50 s^−1^). ** Information given on the product pack. *** Native maize meal flour with unhydrolyzed or non-depolymerized starch molecules.

**Table 2 foods-08-00221-t002:** Definitions of the in-mouth texture attributes used during the evaluation of complementary porridges (CPs) by a trained Temporal Check-All-That-Apply (TCATA) sensory panel (*n* = 10).

No.	TCATA Attribute	Definition
1	Soft	Selected when little force is required to orally process and move around the mouth.
2	Smooth	Selected when the sample is perceived as smooth when squeezed between the palate and tongue [48].
3	Creamy	Selected when the sample is perceived as creamy, with a silky smooth sensation in the mouth [48].
4	Grainy	Selected when grainy particles are perceived in the mouth.
5	Too thick/Semi-solid	Viscosity perception of cooked maize meal pastes 15–20% solids in water. Similar to mashed potato [49].
6	Thick	Viscosity perception of cooked maize meal paste (10–15% solids in water). Selected when the sample is perceived as thick (viscous) as opposed to thin like a fluid [48].
7	Thin	Selected when the sample is perceived as thin andfluid-likeas opposed to thick (viscous).
8	Chewy	Selected when the sample requires a substantial number of chews before it is ready to swallow [50].
9	Sticky	Selected when the sample sticks to the teeth and palate [48].
10	Watery	Selected when the sample was perceived as thin and watery [49].
11	Easy to swallow	Selected when the sample requires little effort (exertion/force) to swallow [51].
12	Difficult to swallow	Selected when the sample requires a lot of effort (exertion/force) to swallow.
13	Slimy	Selected when the sample is perceived as slimy and slippery, a mildly sticky perception on the palate/tongue.
14	Pasty	Selected when the sample has the consistency of a (starch) paste, semi-solid with some stickiness.

The terms “watery” and “too thick” were used to anchor the two extremes of the sensory space for porridge viscosity informed by panel feedback.

**Table 3 foods-08-00221-t003:** The effect of porridge type on the citation proportions for some oral texture attributes at three different moments during evaluation by a trained Temporal Check-All-That-Apply (TCATA) sensory panel (*n* = 10): Data analyzed using Cochran Q test followed by Marascuillo’s pairwise comparison test.

Porridge Type	Beginning (6 s)					Middle (16 s)					End (26 s)			
	Thick	Sticky	Too thick	Pasty	Slimy	Thick	Sticky	Too thick	Pasty	Slimy	Thick	Sticky	Pasty	Slimy
Bambara 10%	0.05 ^a^	0.03 ^ab^	0.00 ^a^	0.03 ^ab^	0.25 ^b^	0.00 ^a^	0.00 ^a^	0.00 ^a^	0.05 ^ab^	0.20 ^b^	0.00 ^a^	0.00 ^a^	0.03 ^ab^	0.15 ^b^
Cowpea 10%	0.08 ^a^	0.13 ^cd^	0.00 ^a^	0.05 ^abc^	0.35 ^bc^	0.05 ^ab^	0.08 ^ab^	0.03 ^a^	0.08 ^abc^	0.20 ^b^	0.03 ^ab^	0.05 ^a^	0.05 ^ab^	0.13 ^b^
Cassava 4%	0.00 ^a^	0.10 ^bcd^	0.00 ^a^	0.03 ^ab^	0.40 ^bc^	0.00 ^a^	0.05 ^ab^	0.00 ^a^	0.00 ^a^	0.25 ^bc^	0.00 ^a^	0.05 ^a^	0.00 ^a^	0.13 ^b^
Cassava 6%	0.08 ^a^	0.18 ^de^	0.00 ^a^	0.15 ^cd^	0.43 ^c^	0.05 ^ab^	0.05 ^ab^	0.00 ^a^	0.05 ^ab^	0.38 ^c^	0.00 ^a^	0.05 ^a^	0.03 ^ab^	0.18 ^bc^
Cassava 10%	0.30 ^b^	0.25 ^e^	0.10 ^b^	0.20 ^d^	0.50 ^c^	0.33 ^f^	0.30 ^c^	0.13 ^b^	0.20 ^c^	0.55 ^d^	0.10 ^c^	0.25 ^b^	0.18 ^cd^	0.25 ^c^
Maize 4%	0.00 ^a^	0.03 ^ab^	0.00 ^a^	0.00 ^a^	0.05 ^a^	0.00 ^a^	000 ^a^	0.00 ^a^	0.00 ^a^	003 ^a^	0.00 ^a^	0.00 ^a^	0.00 ^a^	0.00 ^a^
Maize 8%	0.40 ^b^	0.05 ^abc^	0.05 ^ab^	0.03 ^ab^	0.08 ^a^	0.15 ^cd^	0.03 ^a^	0.05 ^a^	0.05 ^ab^	0.00 ^a^	0.03 ^ab^	0.00 ^a^	0.03 ^ab^	0.00 ^a^
Maize 10%	0.33 ^b^	0.10 ^bcd^	0.25 ^d^	0.13 ^bcd^	0.08 ^a^	0.23 ^de^	0.13 ^b^	0.13 ^b^	0.20 ^c^	0.05 ^a^	0.05 ^b^	0.05 ^a^	0.13 ^bcd^	0.03 ^a^
Sorghum 4%	0.05 ^a^	0.00 ^a^	0.00 ^a^	0.00 ^a^	0.05 ^a^	0.00 ^a^	0.00 ^a^	0.00 ^a^	005 ^ab^	0.05 ^a^	0.00 ^a^	0.00 ^a^	0.03 ^ab^	0.00 ^a^
Sorghum 8%	0.30 ^b^	0.03 ^ab^	0.05 ^ab^	0.00 ^a^	0.03 ^a^	0.10 ^bc^	0.08 ^ab^	005 ^a^	0.08 ^abc^	0.03 ^a^	0.05 ^b^	0.03 ^a^	0.08 ^abc^	0.03 ^a^
Sorghum 10%	0.40 ^b^	0.03 ^ab^	0.18 ^c^	0.15 ^cd^	0.00 ^a^	0.25 ^ef^	0.03 ^a^	0.05 ^a^	0.13 ^abc^	0.03 ^a^	0.03 ^ab^	0.00 ^a^	0.05 ^ab^	0.03 ^a^
OFSP 10%	0.00 ^a^	0.00 ^a^	0.00 ^a^	0.08 ^abc^	0.00 ^a^	0.00 ^a^	0.00 ^a^	0.00 ^a^	0.10 ^abc^	0.03 ^a^	0.03 ^ab^	0.00 ^a^	0.10 ^abcd^	0.03 ^a^
OFSP 16%	0.10 ^a^	0.03 ^ab^	0.00 ^a^	0.05 ^abc^	0.03 ^a^	0.00 ^a^	0.08 ^ab^	0.00 ^a^	0.15 ^bc^	0.03 ^a^	0.00 ^a^	0.05 ^a^	0.20 ^d^	0.03 ^a^
A1 *	0.00 ^a^	0.00 ^a^	0.00 ^a^	0.03 ^ab^	0.03 ^a^	0.00 ^a^	0.03 ^a^	0.00 ^a^	0.10 ^abc^	0.05 ^a^	0.00 ^a^	0.00 ^a^	0.10 ^abcd^	0.03 ^a^

^abcdef^ Citation proportions with different letters within a column represent significant differences among treatments at *p* ≤ 0.05 as analyzed using Marascuillo’s test. Responses were pooled across replicates, and oral processing method (40 TCATA runs × 14 samples) was used to determine the citation proportion values. A1 * is a commercial reference porridge.

**Table 4 foods-08-00221-t004:** Effect of complimentary porridge (CP) type and oral processing (OP) method on citation proportions for texture attributes during the initial (1–10 s) and ending (21–30 s) phases of evaluation by the trained Temporal Check-All-That-Apply (TCATA) panel (*n* = 10). Main effects ANOVA followed by Fisher’s Least Significant Difference (LSD) test for pairwise comparisons.

Porridge-Sample	Oral-Method	Initial: 1–10 s					End: 21–30 s				
		Thick	Too thick	Sticky	Slimy	Pasty	Thick	Too Thick	Slimy	Pasty	Swallow (+)
Maize 4%	Normal	0.00 ^a^	0.00 ^a^	0.02 ^ab^	0.04 ^a^	0.00 ^a^	0.00 ^a^	0.00 ^a^	0.00 ^a^	0.00 ^a^	0.64 ^bcdefghij^
	Up-Down	0.00 ^a^	0.00 ^a^	0.01 ^a^	0.06 ^abc^	0.00 ^a^	0.00 ^a^	0.00 ^a^	0.01 ^a^	0.00 ^a^	0.77 ^ij^
Maize 8%	Normal	0.23 ^def^	0.03 ^ab^	0.05 ^abc^	0.02 ^a^	0.00 ^a^	0.05 ^abc^	0.01 ^ab^	0.00 ^a^	0.00 ^a^	0.76 ^hij^
	Up-Down	0.37 ^g^	0.03 ^ab^	0.07 ^abcd^	0.05 ^ab^	0.05 ^abc^	0.01 ^bc^	0.03 ^ab^	0.00 ^a^	0.05 ^abc^	0.59 ^bcdefgh^
Maize 10%	Normal	0.26 ^defg^	0.11 ^cd^	0.07 ^abcd^	0.07 ^abc^	0.08 ^abcd^	0.04 ^abc^	0.02 ^ab^	0.00 ^a^	0.10 ^abcd^	0.62 ^bcdefghi^
	Up-Down	0.31 ^efg^	0.20 ^e^	0.10 ^bcd^	0.00 ^a^	0.13 ^bcde^	0.01 ^cd^	0.00 ^a^	0.10 ^abcd^	0.18 ^cde^	0.53 ^bcde^
Sorghum 4%	Normal	0.01 ^a^	0.00 ^a^	0.00 ^a^	0.02 ^a^	0.01 ^a^	0.00 ^a^	0.00 ^a^	0.02 ^ab^	0.05 ^abc^	0.65 ^cdefghij^
	Up-Down	0.02 ^a^	0.00 ^a^	0.00 ^a^	0.04 ^a^	0.01 ^a^	0.00 ^a^	0.00 ^a^	0.01 ^a^	0.00 ^a^	0.63 ^bcdefghij^
Sorghum 8%	Normal	0.20 ^cde^	0.04 ^abc^	0.04 ^abc^	0.00 ^a^	0.02 ^a^	0.00 ^a^	0.01 ^a^	0.00 ^a^	0.07 ^abc^	0.56 ^bcdefg^
	Up-Down	0.36 ^g^	0.04 ^abc^	0.04 ^abc^	0.04 ^a^	0.05 ^ab^	0.09 ^bcd^	0.03 ^ab^	0.05 ^abcd^	0.05 ^abc^	0.59 ^bcdefgh^
Sorghum 10%	Normal	0.28 ^defg^	0.08 ^bc^	0.02 ^ab^	0.00 ^a^	0.03 ^a^	0.04 ^abc^	0.01 ^a^	0.00 ^a^	0.00 ^a^	0.67 ^defghij^
	Up-Down	0.32 ^fg^	0.17 ^de^	0.03 ^abc^	0.01 ^a^	0.16 ^de^	0.06 ^abc^	0.02 ^ab^	0.05 ^abcd^	0.11 ^abcd^	0.63 ^bcdefghij^
Bambara 10%	Normal	0.02 ^a^	0.00 ^a^	0.01 ^a^	0.18 ^cd^	0.00 ^a^	0.00 ^a^	0.00 ^a^	0.15 ^bcde^	0.00 ^a^	0.75 _hij_
	Up-Down	0.03 ^a^	0.00 ^a^	0.01 ^a^	0.18 ^cd^	0.05 ^ab^	0.00 ^a^	0.00 ^a^	0.15 ^bcde^	0.05 ^abc^	0.76 ^hij^
Cowpea 10%	Normal	0.07 ^ab^	0.00 ^a^	0.09 ^abcd^	0.24 ^de^	0.01 ^a^	0.00 ^a^	0.00 ^a^	0.11 ^abcde^	0.05 ^abc^	0.70 ^efghij^
	Up-Down	0.05 ^a^	0.00 ^a^	0.09 ^abcd^	0.28 ^def^	0.06 ^abc^	0.07 ^abcd^	0.00 ^a^	0.16 ^cde^	0.08 ^abc^	0.52 ^bcd^
Cassava 4%	Normal	0.00 ^a^	0.00 ^a^	0.08 ^abcd^	0.28 ^def^	0.02 ^a^	0.00 ^a^	0.00 ^a^	0.13 ^abcde^	0.00 ^a^	0.73 ^ghij^
	Up-Down	0.00 ^a^	0.00 ^a^	0.08 ^abcd^	0.35 ^ef^	0.05 ^ab^	0.00 ^a^	0.00 ^a^	0.18 ^de^	0.00 ^a^	0.55 ^bcdef^
Cassava 6 %	Normal	0.05 ^a^	0.00 ^a^	0.12 ^cde^	0.33 ^ef^	0.07 ^abcd^	0.00 ^a^	0.00 ^a^	0.19 ^de^	0.00 ^a^	0.67 ^cdefhij^
	Up-Down	0.06 ^a^	0.00 ^a^	0.16 ^def^	0.39 ^f^	0.14 ^cde^	0.01 ^ab^	0.00 ^a^	0.24 ^de^	0.08 ^abc^	0.60 ^bcdefhi^
Cassava 10%	Normal	0.19 ^bcd^	0.04 ^abc^	0.22 ^ef^	0.38 ^f^	0.08 ^abcd^	0.05 ^abc^	0.01 ^a^	0.24 ^ef^	0.09 ^abc^	0.70 ^efghij^
	Up-Down	0.27 ^defg^	0.17 ^de^	0.26 ^f^	0.35 ^ef^	0.20 ^e^	0.15 ^d^	0.03 ^ab^	0.35 ^f^	0.22 ^de^	0.26 ^a^
OFSP 10%	Normal	0.00 ^a^	0.00 ^a^	0.00 ^a^	0.00 ^a^	0.06 ^abc^	0.00 ^a^	0.00 ^a^	0.03 ^abc^	0.13 ^bcde^	0.56 ^bcdefg^
	Up-Down	0.00 ^a^	0.00 ^a^	0.00 ^a^	0.00 ^a^	0.08 ^abcd^	0.05 ^abc^	0.00 ^a^	0.00 ^a^	0.05 ^abc^	0.65 ^cdefghij^
OFSP 16%	Normal	0.03 ^a^	0.00 ^a^	0.02 ^ab^	0.00 ^a^	0.03 ^a^	0.00 ^a^	0.00 ^a^	0.00 ^a^	0.13 ^abcd^	0.47 ^b^
	Up-Down	0.10 ^abc^	0.00 ^a^	0.05 ^abc^	0.03 ^a^	0.08 ^abcd^	0.00 ^a^	0.00 ^a^	0.05 ^abcd^	0.25 ^e^	0.50 ^bc^
A1	Normal	0.00 ^a^	0.00 ^a^	0.00 ^a^	0.02 ^a^	0.01 ^a^	0.00 ^a^	0.05 ^b^	0.00 ^a^	0.05 ^abc^	0.68 ^defghi^
	Up-Down	0.00 ^a^	0.00 ^a^	0.00 ^a^	0.04 ^a^	0.05 ^abc^	0.00 ^a^	0.00 ^a^	0.05 ^abcd^	0.15 ^cde^	0.72 ^fghi^
A2	Normal	0.08 ^abc^	0.00 ^a^	0.00 ^a^	0.00 ^a^	0.00 ^a^	0.00 ^a^	0.00 ^a^	0.04 ^abc^	0.05 ^abc^	0.62 ^bcdefghi^
	Up-Down	0.06 ^a^	0.00 ^a^	0.00 ^a^	0.00 ^a^	0.04 ^ab^	0.05 ^abc^	0.00 ^a^	0.01 ^a^	0.11 ^abcd^	0.53 ^bcde^
A3	Normal	0.00 ^a^	0.00 ^a^	0.00 ^a^	0.00 ^a^	0.01 ^a^	0.00 ^a^	0.00 ^a^	0.00 ^a^	0.00 ^a^	0.80 ^j^
	Up-Down	0.01 ^a^	0.00 ^a^	0.00 ^a^	0.00 ^a^	0.04 ^ab^	0.00 ^a^	0.00 ^a^	0.00 ^a^	0.02 ^ab^	0.65 ^cdefghi^

For the same column, mean values with different superscripts are significantly different (*p* ≤ 0.05). Values are average citations over a 10 s oral processing period. A1 is a commercial reference. A2 and A3 are selected commercial porridge samples. Swallow (+) refers to easy to swallow.

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
