# Peer review of "Dynamic Oral Texture Properties of Selected Indigenous Complementary Porridges Used in African Communities"

_foods, 2019, doi:10.3390/foods8060221_

Round 1
Reviewer 1 Report
This paper describes the effect of selected complementary porridges (used in African communities) on the oral textural properties using the temporal-check-all-that-apply (TCATA) methodology. The paper introduces the use of this new sensory methodology to measure the textural properties of porridges over the mastication period. Overall, the paper is well written, and the results are consistent with the literature. The purpose of the paper is to understand the sensory textural properties of porridges for the nutrition of infants and young children. As nutrient release is stated as a major factor for malnutrition, this paper needs to discuss how the different textural properties can affect the nutrients release. Other comments can be found below:
Abstract: When describing the results, add a line or two regarding the TCATA results. It can be stated which attributes were expressed at the beginning of the mastication, and which attributes at the end.
Introduction
Line 37 – Remove the word “com”
Lines 60 to 64 – The mechanism behind the effects of CP with a high viscosity (more than 3 Pa.s) on the adequate nutrition of infants should be clearly explained in the introduction section. Is it because of the nutrients release at the OP? Alternatively, mastication and digestibility are factors that can contribute to the nutrients release. It needs more clarification.
Materials and Methods
Line 81 – It seems that different amounts of flour were used for each CP in the mixes. How were these quantities determined? How can these quantities be appropriate for infants in terms of OP and nutritional value?
Table 1 – There should be an explanation in the materials and methods of why the solid contents (%) of the CP and commercial porridge were different (4-16 vs. 24-35%). Different content of solids can affect the viscosity and rheological properties of the porridge. This can have implications on the oral processing of the product.
Author Response
Foods 507872 Response to Reviewer 1 Comments
Comments and Suggestions for Authors
This paper describes the effect of selected complementary porridges (used in African communities) on the oral textural properties using the temporal-check-all-that-apply (TCATA) methodology. The paper introduces the use of this new sensory methodology to measure the textural properties of porridges over the mastication period. Overall, the paper is well written, and the results are consistent with the literature. The purpose of the paper is to understand the sensory textural properties of porridges for the nutrition of infants and young children. As nutrient release is stated as a major factor for malnutrition, this paper needs to discuss how the different textural properties can affect the nutrients release. Other comments can be found below:
Point 1: Abstract: When describing the results, add a line or two regarding the TCATA results. It can be stated which attributes were expressed at the beginning of the mastication, and which attributes at the end.
Response 1: The reviewer’s suggestions were noted. The text in the abstract was changed to describe more clearly the TCATA results, mentioning the attributes that were more relevant during the initial and the final stages of OP. L22-27.
Point 2: Introduction
Line 37 – Remove the word “com”
Response 2: Typo error corrected
Point 3: Lines 60 to 64 – The mechanism behind the effects of CP with a high viscosity (more than 3 Pa.s) on the adequate nutrition of infants should be clearly explained in the introduction section. Is it because of the nutrients release at the OP? Alternatively, mastication and digestibility are factors that can contribute to the nutrients release. It needs more clarification.
Response 3: The comments are well acknowledged. The mechanisms have been explained in L98 - 113 of the introduction.
Point 4: Materials and Methods
Line 81 – It seems that different amounts of flour were used for each CP in the mixes. How were these quantities determined? How can these quantities be appropriate for infants in terms of OP and nutritional value?
Response 4: A sentence was added to explain how the quantities were determined (L133 – 135).
The solids % at which the porridge viscosity was 3 Pa.s at 40 °C and shear rate of 50/s was selected for each flour treatment through a series of factor level experiments.
As mentioned in L. 135 the recommended CP viscosity limit is [3 Pa.s at 40 °C and shear rate of 50 s-1,
The limitation of the CP % solid content as a function of viscosity leading to inadequate nutrient intake is the motivation for the research. See L 95 – 98 and L61 -65.
Good quality complementary food must have low viscosity, high nutrient density, appropriate texture and a consistency that allows for easy consumption by infants and young children (WHO, 2003, Balasubramanian et al., 2014).
Point 5: Table 1 – There should be an explanation in the materials and methods of why the solid contents (%) of the CP and commercial porridge were different (4-16 vs. 24-35%). Different content of solids can affect the viscosity and rheological properties of the porridge. This can have implications on the oral processing of the product.
Response 5: This has been addressed in point 4. See changes L133- 135, 138
The processing techniques for commercial porridges allow them to, remain thin at a higher solids % compared to indigenous porridges.
The flour quantities for each treatment represent the maximum solids % determined from rheological experiments, that gives the recommended baby porridge viscosity of 3 Pa.s (at 40 °C and shear rate of 50 s-1).

Reviewer 2 Report
The presented paper deals with an extremely important and noble topic. The authors aimed to uncover the texture parameters of numerous porridge samples and materials. The presented sample preparation, the used sensory methods and the data analyses are appropriate to fulfill these aims. However, the paper has some structural problems which makes reading difficult and might create a bad feeling in the reader. Since I found the scientific part appropriate, I focused on the formatting problems in my review:
The introduction part presents adequate information about oral food processing, the requirements of complementary foods and the physiological characteristics of children’s mouth. However, I miss some references about how to conduct sensory analysis of infant food products. Which aspects should be taken into account? What training of assessors should be carried out? Are there any standards about it? How is it done by infant food companies? It would be important to cite some relevant papers in order to show the readers that the authors are aware of the sensory methodologies and the best methods were chosen to fulfill the goals of the paper.
Additionally, it would be important to add a short section about preferences. Preferences of infants are complicated to measure. Since mothers taste the food products first (before the infants), it is important to produce food products appealing to the mothers, too. If there is a product, a mother would not taste do to disgust (cause by anything from earlier experiences to aversion), she will not likely to feed her infant with that given product.
I also suggest to start a new paragraph for the aims of the paper to make is easier for the readers.
The sample preparation section is well-written and all necessary information is presented. Table 1 is well-structured.
Please explain, why did you use 10 panelists, add some additional references about the use of TCATA (number of panelist, number of terms etc.). Additionally, the term TCATA is defined only in the abstract.
Some information about tempR package is missing: version of R, version of tempR, citation of R-project, similarly as the authors cited XlStat (XLStat software 154 (version 2019.1.2) (Addinsoft, 2019)(Boston, USA)).
From lines 185 to lines 204, the font size and also the style of the paper changes. The in some cases commas are used as decimals instead of dots and the spaces before units (e.g. %) are missing.
Table 3 the authors use “p” but in Table 3 they use “P”.
Values presented in Table 3 are in German style (commas are used instead of dots), please change.
As I read the paper, this problem with commas and dots are present in many cases, so please go through the paper and correct EACH values.
Why did you change the style of Table 4? The presentation of homogenous subgroups is correct in Table 3 (letters are in superscript) but in Table 4 you inserted a new column for them. Please use the same format as you did in Table 3.
L217. Now you use principal component analysis, however, there is not a single word about PCA in the methods or Data analysis section. The data analysis part is not clear. Additionally, please define PCA. Presentation of Figure 2 is very poor. The four plots (I assume biplots) have different sizes, and they are differently scaled. Why did you put the explained variance values into the title of the plots? These are presented on the axis labels. Furthermore, there is a typo on Fig. 2 D (Diff).
L37: “low cost com complementary” possible typo
L114: facilitate instead of facilitates
L115: panelist instead of panellist.
L124: panelist instead of panellist.
L182: use dot instead of comma
L160: e.g. should be italic.
I miss one important information. The authors evaluated a high number of samples but there are no suggestions to mothers (producers etc.) about which product gave the most satisfactory results, or the list of products meeting the desired criteria. The authors should include this information in the Conclusions section.
After taking into account the above mentioned points, I truly think that the paper is appropriate for publication in Foods.
Author Response
Foods 507872
Response to Reviewer 2 Comments
Comments and Suggestions for Authors
The presented paper deals with an extremely important and noble topic. The authors aimed to uncover the texture parameters of numerous porridge samples and materials. The presented sample preparation, the used sensory methods and the data analyses are appropriate to fulfill these aims. However, the paper has some structural problems which makes reading difficult and might create a bad feeling in the reader. Since I found the scientific part appropriate, I focused on the formatting problems in my review:
Point 1: The introduction part presents adequate information about oral food processing, the requirements of complementary foods and the physiological characteristics of children’s mouth. However, I miss some references about how to conduct sensory analysis of infant food products. Which aspects should be taken into account? What training of assessors should be carried out? Are there any standards about it? How is it done by infant food companies? It would be important to cite some relevant papers in order to show the readers that the authors are aware of the sensory methodologies and the best methods were chosen to fulfill the goals of the paper.
Response 1: We welcome the observation from the reviewer. Background information of sensory analysis with infant food products has been addressed in paragraph 3 of the introduction. The challenges associated with involving infants in sensory research was also explained (L74 -81).
Yet, infants and toddlers present a challenge to sensory and consumer researchers- because of their inability to communicate verbally, limited cognitive abilities, very low attention span [24, 25]. Sensory testing with infants and young children therefore often employ indirect approaches.. Descriptive sensory profiling have been used to evaluate the sensory quality of baby foods (purees) [4].
Point 2: Additionally, it would be important to add a short section about preferences. Preferences of infants are complicated to measure. Since mothers taste the food products first (before the infants), it is important to produce food products appealing to the mothers, too. If there is a product, a mother would not taste due to disgust (cause by anything from earlier experiences to aversion), she will not likely to feed her infant with that given product.
Response 2: The author acknowledges the comment. While this particular work focussed of profiling the temporal in-mouth texture of baby porridges in light of the limited but constantly evolving oral processing abilities of young children, preference and acceptance issues in baby foods are also highly relevant. These were addressed in paragraph L83 - 93 in the introduction.
For preference evaluation, parents’ liking is important in deciding if a given CP would be suitable for their infants [26]. The primary caretaker (typically the mother) interprets the behaviour of the child during food tasting and rates acceptance on a hedonic scale [27, 28, 11]. Having the adult also taste and rate the samples after the child provides a control and confirmatory of the acceptability of the samples [25]. Mothers were asked to report on presence/absence of positive and negative behaviours, and on infant's liking during feeding. Alternative testing approaches employed include parents completing an Infant Behaviour Questionnaire, and rating of videotapes of infants’ facial reactions to foods rated [29].
Point 3: I also suggest to start a new paragraph for the aims of the paper to make is easier for the readers.
Response 3: Thank you for the suggestion. The recommendation was implemented.
Point 4: Please explain, why did you use 10 panelists, add some additional references about the use of TCATA (number of panelist, number of terms etc.). Additionally, the term TCATA is defined only in the abstract.
Response 4: The motivation for use of 10 panelists and 14 attributes has been explained in L169 -177.
TCATA has often been used with a large number of consumers in evaluating different products [43, 44]. However coupled with training, fewer panellists (10 – 15) have also previously been used for th temporal profiling of products based on 8 – 10 attributes [41, 45]. The current study used 10 panelists well trained in carrying out the TCATA task and a list of 14 attributes. Temporal methods are more cognitively demanding, usually rely on shorter lists [46, 47].
Point 5: Some information about tempR package is missing: version of R, version of tempR, citation of R-project, similarly as the authors cited XlStat (XLStat software 154 (version 2019.1.2) (Addinsoft, 2019)(Boston, USA)).
Response 5: Referencing for R and tempR as well as versions have been corrected for. L212 -214 in section 2.4.
Point 6: From lines 185 to lines 204, the font size and also the style of the paper changes. The in some cases commas are used as decimals instead of dots and the spaces before units (e.g. %) are missing.
Response 6: Font size, style, spaces, and comma issues were corrected.
Point 7: Table 3 the authors use “p” but in Table 3 they use “P”.
Values presented in Table 3 are in German style (commas are used instead of dots), please change.
Response 7: The author kindly takes note of the errors highlighted. A p small cap has been adopted in use consistently. Values in Table 3 changed.
Point 8: As I read the paper, this problem with commas and dots are present in many cases, so please go through the paper and correct EACH values.
Response 8: This was also Corrected throughout the paper.
Point 9: Why did you change the style of Table 4? The presentation of homogenous subgroups is correct in Table 3 (letters are in superscript) but in Table 4 you inserted a new column for them. Please use the same format as you did in Table 3.
Response 9: Table 4 has been reformatted, columns for homogeneous subgroups removed, and superscript format of letters used.
Point 10: L217. Now you use principal component analysis, however, there is not a single word about PCA in the methods or Data analysis section. The data analysis part is not clear. Additionally, please define PCA. Presentation of Figure 2 is very poor. The four plots (I assume biplots) have different sizes, and they are differently scaled. Why did you put the explained variance values into the title of the plots? These are presented on the axis labels. Furthermore, there is a typo on Fig. 2 D (Diff).
Response 10: Details of the PCA was included as suggested in section 2.4 Data analysis as follows:
Principal Component Analysis (PCA)- a multivariate data analysis method for and visualization of correlations between multiple quantitative observations and variables [58], was used to produce PCA product trajectories (biplot). Visualizing observations in a 2- or 3-dimensional space permits identification of uniform or atypical groups of observations. PCA can be considered as a projection method which projects observations from a p-dimensional space with p variables to a k-dimensional space (where k < p) so as to conserve the maximum amount of information from the initial dimensions [58].
Figure 2 has been improved as suggested.
Point 11: L37: “low cost com complementary” possible typo
L114: facilitate instead of facilitates
L115: panelist instead of panellist.
L124: panelist instead of panellist.
L182: use dot instead of comma
L160: e.g. should be italic.
Response 11: The typo error and spellings have been corrected accordingly.
Point 11: I miss one important information. The authors evaluated a high number of samples but there are no suggestions to mothers (producers etc.) about which product gave the most satisfactory results, or the list of products meeting the desired criteria. The authors should include this information in the Conclusions section.
Response 11: CP recommendations (OFSP and legumes) here have been made mainly in terms of oral texture, considering that these CPs at these porridges were not so much profiled as thick, too thick or sticky at the highest solids content under this investigation. Recommendations have been made for industry in terms of how they can use the results to produce infant foods that are optimized for sensory quality and nutrition. L478 – 494
OFSP porridge had satisfactory oral texture at its highest solids content, comparable to a commercial reference (A1). Parents and caregivers are advised to consider the use of OFSP flour in composite with a legume (e.g. Cowpea or Bambara) for the preparation of CPs with relatively high solids content, suitable oral texture and nutritive quality. Simple traditional approaches for reducing viscosity of indigenous CPs at home (e.g. malting, fermentation, souring) need consideration by primary caregivers. This study provides scientific insight for baby foods manufacturers on the OP characteristics of complimentary foods for infants and young children in African communities. Smart tech innovations for processing indigenous flours that give CP of optimal oral texture and solids content are required. Moreover, further research is needed to explore the dynamic sensory interplay between bolus properties of the foods and the oro-tactile phenomena (tongue coordination, mastication and lubrication) in infants and young children, which at present is not well understood.

Reviewer 3 Report
This manuscript treats oral texture of porridge samples provided to African young children to improve them to prevent malnutrition. Trained panel (10 young adults) evaluated sensory characteristics of 13 porridge samples and 23 commercial products for 6 to 24 months children by TCATA. Young children and adults definitely do different oral processing, though trained panel performed simulated munching and normal chewing, no evidence is presented the simulated method by adults and common children sense similar texture. Even though trained panelists could perform similar movement to children, sensory attributes (soft, smooth, etc.) are likely perceived differently between adults and children due to different size of oral organs, saliva secretion, etc. Moreover, there may be differences between 6- and 24-month children.
The effects of size are not considered at all. Besides body size of adults and children, thicker foods may result smaller bite size within a panelist of normal oral processing. To ingest enough nutrition, total amount of foods is more important than sensory characteristics of a spoonful food. Textural attributes and fullness or satiety during eating of meal size porridge may be different from evaluated texture for a spoonful porridge.
Differences may cause by forced eating behavior to simulate munching, and by order as all panelists tried munching first and then chewing.
Discussion on texture without rheological data seems difficult. I recommend to add some viscosity data.
This manuscript is not prepared according to the Foods template. References must be numbered in order of appearance in the text and listed individually at the end of the manuscript.
Is “indigenous/local” in Table 1 right term?
Periods not commas are used for decimal numbers (Tables 1, 2, L148, L182, L187, L190, L192, L195).
Figures 1 and 2 with small letters and many lines are difficult to read. I could not review these graph data.
Author Response
Foods 507872
Response to Reviewer 3 Comments
Comments and Suggestions for Authors
This manuscript treats oral texture of porridge samples provided to African young children to improve them to prevent malnutrition. Trained panel (10 young adults) evaluated sensory characteristics of 13 porridge samples and 23 commercial products for 6 to 24 months children by TCATA.
Point 1: Young children and adults definitely do different oral processing, though trained panel performed simulated munching and normal chewing, no evidence is presented the simulated method by adults and common children sense similar texture. Even though trained panelists could perform similar movement to children, sensory attributes (soft, smooth, etc.) are likely perceived differently between adults and children due to different size of oral organs, saliva secretion, etc. Moreover, there may be differences between 6- and 24-month children.
Response 1: The comment of the reviewer is noted. The motivation for using adults for the study is presented in L 118 – 119. The limitations as identified by the reviewer are acknowledged. Text was added based on the reviewer’s comments?
L 51 – 61 The process of bolus formation is under the coordinated action of mastication (reduction of food in particles), salivation (lubrication of particles) and tongue movements (agglomeration of particles with saliva and swallowing) [1] and depends thus mainly on the development of the infant masticatory apparatus. For ingestion and break-down of solid foods, children need to acquire specific feeding skills which require more effort than the oral manipulation of liquid milk [14]. The acceptance of food with a given texture, (in this context defined as the infant’s ability to swallow the food [11]), strongly depends on acquisition of feeding skills, which can develop differently in children of the same age [15]. At 12 months, munching/chewing behaviour is well-established and continues to develop and optimize by 2 – 3 years [16]. However, the age of chewing maturation (i.e., transition of up and down movements of the jaw to rotary movements) remains not so clear- estimated to be later than 3 years old [14].
A trained sensory panel consisting of adults was used because infants are not capable of carrying out evaluation instructions and tasks expected in descriptive sensory evaluation.
L377 – 379. Young children and adults differ in oral processing and the use of adults with different oral processing apparatus, saliva secretion etc. cannot fully explain the texture perception of small children.
Point 2: The effects of size are not considered at all. Besides body size of adults and children, thicker foods may result smaller bite size within a panelist of normal oral processing.
Response 2: The limitation identified is noted and acknowledged as per point 1.
Point 3: To ingest enough nutrition, total amount of foods is more important than sensory characteristics of a spoonful food. Textural attributes and fullness or satiety during eating of meal size porridge may be different from evaluated texture for a spoonful porridge.
Response 3: The opinion of the reviewer is noted. In this paper however the focus was on the oral processing of indigenous CPs that are commonly fed to infants and young children in African communities, an aspect that has not been studied enough. As mentioned in the first sentence L32 – 34.
Food oral processing (OP), the manipulation and break down of food inside the mouth up to the moment of swallowing [1, 2], plays a key and important role in sensory perception, consumer acceptance and food intake [3].
Point 4: Differences may cause by forced eating behavior to simulate munching, and by order as all panelists tried munching first and then chewing.
Response 4: The author acknowledges this as a limitation of the study, which may introduce bias due to the “order effects” and the limitation was added (L411 -413).
A limitation of the study is that the panel always applied the Up-Down method first followed by the Normal OP method, which could potentially affect results through order effects.
Point 5: Discussion on texture without rheological data seems difficult. I recommend to add some viscosity data.
Response 5: This observation is kindly appreciated by the authors. A paragraph on rheology has been added to introduction. Rheology literature was also appealed to in the results discussion. However, more rheological data was not included as it is part of a rheology paper that will be submitted for published soon L98 - 110.
Data from rheological studies (not included in this paper) has shown that indigenous porridge samples at very low solids content (Maize 8.1 %, Sorghum 8.4 %, Cassava 6.4 %, Bambara groundnut 10.7 %, and Cowpea 10.1 %), exceeded the recommended CP viscosity limit [3 Pa.s at 40 °C and shear rate of 50 s-1, [30, 31]] for infants and children below 3 years of age. When starch is heated in water it swells, gelatinizes and pastes to form a thick gruel [8-10]. Infants and young children have difficulty to consume and swallow aviscous porridge due to their limited oromotor capacity [32].The thickness, or viscosity of shear-thinning foods is perceived by mechano-receptors in the mouth, and depend on the in-mouth shear stress applied and the resultant shear rate [33]. At the critical solids concentration (c*) the gelatinized, amorphous random starch polymer coils come in contact with one another, eventually overlapping at the entanglement concentration (ce) [34]. Polymers with larger molecules display effective molecular entanglements [35, 36] and are generally perceived as more viscous and thick. High viscosity in CP elicits highlingual swallowing pressure [37]. Thicker and harder foods are eaten at a slower rate, often requiring smaller bite sizes, and more chewing time in the mouth before swallowing compared to softer foods [38]. The formation of a bolus that can be safely swallowed is a complex oral process [11], and infants and young children have limited oral capacity to perform this process.
Point 6: This manuscript is not prepared according to the Foods template. References must be numbered in order of appearance in the text and listed individually at the end of the manuscript.
Response 6: Referencing has been corrected. Changes in paper format corrected.
Point 7: Is “indigenous/local” in Table 1 right term?
Response 7: The authors would like to retain this description, considering that some of the samples are strictly indigenous to African communities while some have become the staple foods locally (e.g maize) although they originated out of Africa.
Point 8: Periods not commas are used for decimal numbers (Tables 1, 2, L148, L182, L187, L190, L192, L195).
Response 8: Commas were replaced with periods.
Point 9: Figures 1 and 2 with small letters and many lines are difficult to read. I could not review these graph data.
Response 9: The diagrams for figures 1 and 2 have been improved and the presentation format changed to enhance a better view.

Round 2
Reviewer 3 Report
The authors responded to the previous comments. Introduction and discussion, and style have been appropriately amended.
However, I do not think that Figures 1 and 2 have been appropriately modified. There are small letters in Fig. 1, and many color lines and overlaid letters in Fig. 2. I must repeat that I could not read these graph data.
Author Response
The authors responded to the previous comments. Introduction and discussion, and style have been appropriately amended.
All corrections made on the previous submitted version were accepted in this new revised submission. The document was check for formatting.
However, I do not think that Figures 1 and 2 have been appropriately modified. There are small letters in Fig. 1, and many color lines and overlaid letters in Fig. 2. I must repeat that I could not read these graph data.
The small letters in Fig 1 were corrected
Fig 2 were redone to facilitate easier reading.
